# Effects of Microplastics and Nanoplastics Exposure on Neurogenesis: Are Thymidine Analogs a Good Option to Study Such Effects?

**DOI:** 10.3390/ijms26167845

**Published:** 2025-08-14

**Authors:** Mercè Encinas, Joaquin Martí Clúa

**Affiliations:** Unidad de Citología e Histología, Departament de Biologia Cel·Lular, de Fisiologia i d’Immunologia, Facultad de Biociencias, Institut de Neurociències, Universidad Autónoma de Barcelona, 08193 Bellaterra, Barcelona, Spain; merce.encinas@uab.cat

**Keywords:** microplastics, nanoplastics, central nervous system, neurogenesis, thymidine analogues, BrdU-labeling

## Abstract

An important disadvantage of plastics is their fragmentation into smaller particles, classified according to size as microplastics and nanoplastics. These plastic particles persist for extended periods in aerial, terrestrial, and aquatic ecosystems and can be incorporated into animal bodies through various routes, including inhalation, dermal contact, and the food chain. The accumulation of these debris generates toxicity on several organs, including the nervous system. In this review article, I will cover the detrimental consequences of plastic exposure on the nervous system, the impact of microplastics and nanoplastics on the genesis of neurons both in the embryonic period as well as in adulthood, and the reliability of 5-bromo-2′-deoxyuridine (BrdU) labeling as a tool to analyze the effect of microplastic and nanoplastic exposure on the proliferative behavior of neuronal precursors. BrdU is a marker of DNA synthesis. It is widely used to identify proliferating neuroblasts and follow their fate during embryonic, perinatal, and adult neurogenesis. However, the use of BrdU labeling for analyzing neurogenesis may be inaccurate due to pitfalls and limitations. This is because BrdU exposure can induce apoptosis, cellular senescence, and alterations in DNA methylation. Interestingly, these cellular events also occur following exposure to plastic particles.

## 1. Introduction

Plastic is a word that originally meant pliable and easily shaped. The first plastic was developed in the mid-19th century by John Wesley Hyatt as a substitute for ivory. Since then, plastics have been widely used in modern industry around the world. Nowadays, almost all aspects of daily life involve plastics. They have revolutionized industries such as food packaging, construction, medical equipment, and electronics [1]. According to the International Union of Pure and Applied Chemistry, plastic is defined as a polymeric material. There are seven major plastic species: polypropylene, high- and low-density polyethylene, polyvinyl chloride, polyethylene terephthalate, polystyrene, and polyurethane. They are produced from nonrenewable petroleum and natural gas [2]. To these polymers, several additives are added to impart or enhance some physical and chemical properties, including resistance to solar irradiation, ductility, and color.

The use of plastic goods presents both advantages and disadvantages. Among their advantages are their low cost, light weight, durability, and resistance to mechanical damage, corrosion, and adverse weather conditions. However, their disadvantages mainly arise from overproduction and inadequate recycling practices. In 2022, global plastic production exceeded 400 million tons. If this current trend persists, it is presumable that plastic production will triple, and about 12 billion metric tons will be released into the environment by 2050 [3]. Several studies have revealed that only 18% of plastics are recycled, and 24% are incinerated. The remaining 58% is returned to the environment as waste, where they persist for a long time [4,5]. It is unknown how long plastic waste has been present in the ecosystems. It is believed that, due to their slow degradation, plastics may persist for centuries or even millennia [6].

Other major disadvantages of plastics are their additives and fragmentation. There are several molecules classified as plastic additives, including stabilizers, fillers, and cross-linking agents. These chemicals can accumulate within host organisms, resulting in adverse biological effects [7]. In this context, it has been observed that phthalates are related to low fertility in humans and premature development in women. Similarly, bisphenol A and polybrominated diphenyl ethers are associated with endocrine disruption and respiratory toxicity [8].

On the other hand, the breakage of plastics into small debris is another serious issue in our society. Plastic fragments are categorized into macroplastics (fragments larger than 5 mm), microplastics (particles smaller than 5 mm), and nanoplastics (particles smaller than 0.1 µm). The latter two are generated either through primary industrial processes or via the degradation of macroplastics by various physical, chemical, and biological mechanisms [9]. Due to their small size and physicochemical properties, microplastics and nanoplastics become bioavailable and have the potential to bioaccumulate. These persistent and ubiquitous contaminants appear in high concentrations in terrestrial and aquatic ecosystems, including oceans, soils, and plants [10,11]. Microplastics and nanoplastics can serve as carriers for several environmental contaminants, including organic pollutants, antibiotics, and pathogenic microorganisms [12,13]. Furthermore, plastic debris has detrimental effects on both plants and animals. In animals, plastic particle accumulation occurs through inhalation, dermal exposure, and ingestion of contaminated foodstuffs of both animal and vegetable origin [14,15]. After exposure, microplastics and nanoplastics are found in numerous human biological specimens, including plasma of blood, stools, saliva, and mother’s milk, as well as in the lung, liver, and brain. Moreover, previous studies have indicated that accumulation of these pollutants produces toxicity in different tissues and organs of invertebrates and vertebrates, including the gastrointestinal tract, liver, kidney, lung, and nervous system [11,15,16,17].

In the ambit of neurogenesis, a pivotal issue in both embryonic and adult neurogenesis research is the reliable identification, in histological sections, of those neuroblasts engaged in DNA synthesis [18]. The possibility of tagging proliferative neuron precursors and dating timetables of neurogenesis began with the application of tritiated thymidine and subsequent autoradiography in fixed samples [19,20]. This procedure is arduous, requires technical expertise to ensure the handling of samples, and needs a well equipped laboratory. These limitations were overcome with the use of thymidine analogs. Several pyrimidine analogs have been used: 5-bromo-2′-deoxyuridine (BrdU), 5-chloro-2′-deoxyuridine, 5-iodo-2′-deoxyuridine, and 5-ethynyl-2′-deoxyuridine. Recently, novel nucleotide analogs such as (2′S)-2′-deoxy-2′-fluoro-5-ethynyluridine, 5- (azidomethyl) -2′-deoxyuridine, and 5-vinyl-2′-deoxyuridine have been developed. These analogs share a common feature: all incorporate into nascent DNA [21]. Of all these, BrdU is the most widely used. This synthesized bromide-labeled base analog is chemically and structurally distinct from thymidine. BrdU incorporates a completely foreign atom (Br) into replicating DNA by substituting bromouracil for thymidine [18]. Moreover, this halopyrimidine crosses the blood–brain barrier (BBB) via nucleoside transporters and is permanently integrated into newly synthesized DNA during the S-phase [21,22]. Once incorporated into nascent DNA, BrdU remains stable for prolonged periods of time, and it will be passed down to daughter neuroblasts following the mitotic phase of the cell cycle [23].

In this review, I will cover the following items: (i) the harmful effects of plastic particle exposure on the central nervous system (CNS), (ii) the impact of microplastics and nanoplastics on the genesis of neurons both in the embryonic period as well as in adulthood, and (iii) the reliability of BrdU labeling as a tool to analyze the effect of microplastic and nanoplastic exposure on the proliferative behavior of neuronal precursors. In this section, I will also demonstrate that both plastic debris and BrdU induce apoptosis, cellular senescence, and alterations in DNA methylation. Therefore, the use of this marker may not be appropriate for assessing the effects of plastic particles.

## 2. Harmful Effects of Plastic Particle Exposure on the Central Nervous System

Plastic particles are found in aquatic, terrestrial, and aerial areas. These debris produce cardiovascular, hepatic, and renal injury [24,25]. The nervous system is also affected. In this context, previous research has provided evidence of the transplacental transfer of plastic particles to the fetal brain following exposure of pregnant dams. Plastics were found in the cerebellum, hippocampus, striatum, prefrontal cortex, hippocampus, substantia nigra, and pituitary. The cerebellum, hippocampus, striatum, and prefrontal cortex were the regions containing the highest accumulation of plastic particles [26]. In addition, plastic particles have been shown to cross the BBB and can subsequently accumulate in both gray and white matter.

An in vivo study revealed that plastic particles are found in the mouse brain 120 min after exposure [11]. The mechanisms by which microplastics and nanoplastics cross the BBB are not yet fully understood. At least two mechanisms have been proposed: The first of them has proposed that plastic debris increases the permeability of the BBB through the impairment of the zonulae occludentes located in the endothelial cells [10]. The second mechanism, using computer models, suggests that the presence of a biomolecular corona around plastic particles is needed for their ability to cross the BBB. The second mechanism, using computer models, suggests that the presence of a biomolecular corona around plastic particles is needed for their ability to cross the BBB. The same study indicates that cholesterol molecules promote the ability of plastic particles to cross the BBB [11].

When located in the brain, plastic debris produces deleterious effects through several mechanisms such as: (i) oxidative stress, (ii) inhibition of acetylcholinesterase, (iii) mitochondrial malfunction, and (iv) inflammatory reactions. In the first mechanism, it has been reported that exposure to microplastics and nanoplastics in mice increases the production of reactive oxygen species and malondialdehyde, while simultaneously decreasing glutathione levels in the CNS [27,28]. The same study also demonstrated that learning and memory activity were altered in the Morris water maze paradigm [27]. Interestingly, excessive levels of reactive oxygen species are negatively related to alterations in neural development during perinatal life [29]. Another experiment has reported that plastic particle exposure promotes alterations in the activities and levels of superoxide dismutase, catalase, and malondialdehyde in the brain of the *Oryzias javanicus*, which suggests that plastic debris generates an imbalance between the production of reactive oxygen species and the capacity of the cellular antioxidant system to ameliorate that production [30]. In the annelid *Eisenia foetida*, on the other hand, it has been reported that plastic debris produces modifications in the content of malondialdehyde as well as in the catalase activity [31].

Acetylcholinesterase is a cholinergic enzyme involved in the breakdown of acetylcholine into choline and acetate. Because acetylcholine is a neurotransmitter that regulates motor neuron function of the gray matter, the inhibition of acetylcholinesterase activity is a signal of neurotoxicity [32]. Several studies have reported that exposure to microplastics and nanoplastics inhibits acetylcholinesterase activity in various marine organisms, a process associated with the disruption of normal CNS [30,33,34]. The effects of plastic particles on acetylcholinesterase activity are not limited to aquatic species but have also been observed in terrestrial mammals. In this context, treatment with plastic particles has been shown to alter the activity of this hydrolase in both the liver and the brain [35].

Another important issue related to microplastic and nanoplastic exposure is the effect of these pollutants on mitochondrial function. This cytoplasmic organelle is involved in critical cellular functions, including aerobic respiration to generate ATP. The effects of plastic particle exposure on neuronal mitochondrial function have been shown to depend on particle size. Several studies have indicated that nanoplastics exert greater effects than microplastics, likely due to their higher propensity to accumulate in both the inner and outer mitochondrial membranes [36,37]. In this context, exposure to plastic particles has been found to impair mitochondrial oxidative phosphorylation in dopaminergic-differentiated SH-SY5Y cells, leading to dysfunctions in mitochondrial membrane potential, oxygen consumption rate, and ATP production via the AMPK/ULK1 pathway [38]. Consistent with these findings, nanoplastic exposure in rodents has been associated with loss of Nissl substance and neuronal depletion in the pars compacta of the substantia nigra and the striatum, attributed to mitochondrial dysfunction [39]. Moreover, in zebrafish neurons, plastic particle exposure has been shown to cause mitochondrial impairment, which is magnified due to the low copy number of mitochondrial DNA and the expression of the mRNA of genes associated with the inhibition of mitochondrial fusion, activation of the mitochondrial division and mitophagy, and reduced copy number [40].

Exposure to microplastics and nanoplastics has also been associated with inflammatory reactions. In vitro studies using the human microglial HMC-3 cell line have shown that plastic particles alter the expression of immune cell clusters, while in vivo experiments in mice have demonstrated that exposure leads to changes in the expression of microglial differentiation markers, accompanied by activation of the NF-κB pathway and increased levels of pro-inflammatory cytokines in the hippocampus, cerebral cortex, and cerebellum [41]. Furthermore, in an Alzheimer’s disease mouse model (APP/PS1 double-transgenic mice), nanoplastic exposure exacerbates cognitive impairment and promotes neuroinflammation [42]. Moreover, administration of nanoplastics—but not microplastics—has been shown to induce overexpression of glial fibrillary acidic protein (GFAP) in the cornu ammonis 3 of the hippocampus [43]. In addition, only nanoplastics have been reported to cause a shift in quiescent microglia to a reactive phenotype in both the cerebral cortex and the CA3 region of the hippocampus, leading to the loss of Nissl bodies and neuronal injury [10,43].

## 3. Detrimental Consequences of Plastic Particle Exposure on the Production of Neurons Both in the Prenatal and Adult Life

During the embryonic and perinatal development of the CNS, neural stem cells produce astrocytes and oligodendrocytes as well as several types of neurons. In this specialized process, neural stem cells originate from neuroepithelial cells located in the neural tube [44]. Neurons are produced from prenatal life until early postnatal stages, with only a few neurogenetic regions remaining active in adulthood. In most adult mammals, neural stem cells are found in two brain regions: (i) the dentate gyrus in the hippocampus, where newly produced granule cells move a short distance from the subgranular cell layer to the granular cell layer, and (ii) the ventricular–subventricular area lining the lateral ventricles, where newly generated neurons migrate to the olfactory bulb via the rostral migratory stream. Neural stem cells, derived from embryonic radial glia, are involved in the generation of neurons, astrocytes, and oligodendrocytes [45,46,47].

As indicated above, a transplacental transfer of plastic particles to the fetal brain has been demonstrated following maternal exposure. Furthermore, previous studies have shown that plastic particles can cross the BBB and accumulate in neuroblasts, neurons, glial cells, and endothelial cells. Once located in the brain parenchyma, microplastics and nanoplastics exert deleterious effects in both invertebrate and vertebrate species. It has been revealed that plastic particles alter prenatal and adult neurogenesis (Figure 1). For instance, exposure of sea urchins to microplastic particles during embryonic and larval stages resulted in an important depletion of serotonergic sensory neurons in the apical organ of these echinoderms [48]. Similarly, in zebrafish, microplastic exposure during both embryonic and larval stages decreased the expression of two key neurodevelopmental genes: *neurogenin1* (involved in specifying neuronal differentiation) and *olig2* (involved in oligodendrocyte specification and differentiation). Additionally, the same study also reported a reduction in the number of PCNA (involved in the control of DNA replication)—positive cells and SOX2 (involved in maintaining stem cell pluripotency)—positive cells, indicating impaired proliferative activity of neural progenitors and a reduction in the neural stem cell pool [49].

On the other hand, the effects of microplastics and nanoplastics are also observed in the perinatal period. In proliferating neural stem cells cultured from a 5-day-old mouse subventricular zone and exposed to plastic particles, a substantial reduction in the number of phosphorylated histone H3 and 5-ethynyl-2′-deoxyuridine-reactive cells exists. The cytotoxic effects also affected the generation of neurons and oligodendrocytes [50]. Furthermore, perinatal exposure of mice to nanoplastics leads to neuronal depletion in the cornu ammonis regions 1 and 3, as well as impaired neurogenesis in the dentate gyrus. Interestingly, this effect was not seen after microplastic treatment. Despite that, both microplastics and nanoplastics modify the dendritic spines morphology in the cornus ammonis 1, but only microplastics reduce the spines density in this area of the hippocampus [43].

In another set of experiments, it was demonstrated that proliferating neuronal precursors are vulnerable to the effects of nanoplastic exposure. Consistent with this, in vitro studies have shown that the negative impact of plastic particles on the proliferative dynamics of C17.2 neuronal progenitor cells results from G1 cell cycle arrest. Additionally, nanoplastic treatment increased the expression of cell cycle inhibitors such as p21 and p27, while decreasing the expression of cyclin D. The study also reported that nanoplastic exposure reduced neural stem cell proliferation in the dentate gyrus of the hippocampus, leading to impaired neurogenesis in adult mice and decreased expression of the neural stem cell marker, nestin [28].

In the ventricular–subventricular area lining zone, a transient increase has been observed in the number of proliferating neural stem cells, proliferating neuroblasts, and doublecortin-positive cells [51]. The same study also reported that new interneurons migrate from the ventricular–subventricular zone to the olfactory bulb via the rostral migratory stream.

## 4. Plastic Particles Exposure and Neurogenesis in the Context of the BrdU-Labeling

Plastic particles are found in all ecosystems. These environmental pollutants can enter the animal body following three routes: skin (cosmetics and textiles), inhalation (exposure to textiles, synthetic rubber tires, and plastic covers), and ingestion (consumption of contaminated plants and animals, and products such as toothpaste, beer, and mineral water contained in plastic bottles) [52,53]. The accumulation of these pollutants in the CNS produces a myriad of negative effects, including apoptosis, senescence, and alterations in DNA methylation. The same cellular events are also produced due to BrdU exposure.

### 4.1. Plastic Particles and 5-Bromo-2′-Deoxyuridine Induce Apoptosis

Several studies have indicated the induction of apoptotic cellular events in nerve cells following plastic particle exposure. In this regard, it has been shown that exposure of adult zebrafish to 2 mg/L of plastic particles increases the expression of the apoptotic genes, *caspase-3*, *caspase-9*, and *caspase-8*, in the brain [54]. Furthermore, it was also revealed that mixed neuronal populations isolated from the prenatal mouse cortex present increased expression of cleaved caspase-3 protein after a two-day exposure to 100 mg/L of nanoplastics [55]. Similarly, exposure of cortical neuron cultures to 1 mg/L or 10 mg/L of plastic particles led to a significant increase in cleaved caspase-3-immunoreactive neurons [56]. In addition, exposure to polystyrene nanoparticles for 24 h or 48 h induces apoptosis in mouse Neuro-2a neuroblastoma cells and human HLA-G-positive choriocarcinoma cells. In the same study, a significant increase in apoptotic thalamic neurons was seen in C57BL mouse fetuses following maternal exposure to 1 mg/day of polystyrene nanoparticles administered via intragastric gavage for 17 consecutive days [57].

An important aspect to consider is the relationship between apoptosis and cell proliferation following plastic particle exposure. It has been reported that polystyrene microplastics can simultaneously induce both apoptotic events and cell proliferation. In other words, the activation of apoptosis may trigger a compensatory proliferative response in surrounding healthy cells to maintain tissue homeostasis [56]. In this context, it has been shown that pro-apoptotic caspases can activate the c-Jun N-terminal kinase signaling pathway, which promotes the proliferation of neighboring cells to replace those lost through apoptosis [58]. Similarly, it has been reported that embryonic brain damage induces the proliferation of neural precursors via an apoptosis-induced mechanism [59].

Since the introduction of the first monoclonal antibody against BrdU [60], several antibodies are commercially available [21,22]. This has led to the development of several immunocytochemical methods for detecting BrdU incorporated into replicating DNA. It has provided valuable insights into the cellular mechanisms involved in embryonic and perinatal development of the CNS. These procedures have been used to analyze various processes, including cell cycle kinetics, developmental timetables, and migration and cell lineage in a wide range of species, including mammals [61,62]. Despite these advantages, the possibility of false BrdU labeling or the incorrect interpretation of such labeling should be considered. In this context, the controversy is generated when BrdU labeling can be explained by processes unrelated to cell proliferation, i.e., when a neuroblast is undergoing an apoptotic cellular event.

During embryonic and perinatal development, the neuronal types that populate the nervous system are produced according to strict neurogenetic timetables. These times of neuron origin can be traced via the administration of BrdU [63,64]. In some cases, an excessive number of neurons is generated, leading to the formation of incorrect connections with their target cells. Subsequently, a proportion of these neurons undergo apoptosis as a homeostatic mechanism to adjust their final numbers [65,66]. Cell death is also a normal process in the adult neurogenetic regions, such as the dentate gyrus of the hippocampus and the ventricular–subventricular area, where a significant proportion of newly originated neurons undergo apoptosis before reaching maturity [67,68].

Neurons are fully differentiated cells. They are in the G_0_ phase of the cell cycle. However, several studies have reported that, in response to damage, neurons initiate DNA synthesis without completing cytokinesis [69,70]. In line with these findings, it has been indicated that homocysteine exposure in cultured rat cerebral cortex neurons induces apoptotic cellular events, triggers DNA synthesis, and leads to BrdU incorporation into neuronal nuclei [71]. On the other hand, it has been reported that, following cerebral hypoxia/ischemia, apoptotic neurons in the cornus ammonis 1 region of the hippocampus can reinitiate DNA synthesis and incorporate BrdU into their nuclei [72]. These findings reveal that, after injury, the incorporation of BrdU into a nucleus is not evidence of neurogenesis because BrdU labeling can also occur when nerve cells are undergoing an apoptotic cellular event. In addition to that, it has been reported that a single injection of BrdU at doses ranging from 100 to 300 µg/g leads to the activation of apoptotic events in the rat embryonic cerebellum [73,74].

Collectively, these studies suggest that, in the absence of appropriate controls, BrdU-labeling may not be an appropriate tool for assessing the effects of plastic particles exposure on neurogenesis.

### 4.2. Plastic Particles and 5-Bromo-2’-Deoxyuridine Induce Senescence

Senescence is a dynamic multistep physiological process through which the cell cycle is irreversibly arrested. This cellular event is a protective mechanism to maintain homeostasis and avoid the replication of aged or injured cells that are under certain stress conditions [75]. Cellular senescence can occur at any stage of life, from prenatal period to adulthood [76]. However, the accumulation of senescent cells has been related to several neurological diseases, including amyotrophic lateral sclerosis, Parkinson’s disease, and Alzheimer’s disease. In this context, the senescence in glial cells, neurons, and neural stem cells has been related to these neurodegenerative disorders [42].

The widespread use of plastic products has led to an alarming increase in microplastics and nanoplastics in the environment. These debris can cross biological barriers and induce premature senescence in various cell types and organs. For instance, Wang et al. [77] reported that plastic particles induce senescence in cardiomyocytes by causing mitochondrial oxidative stress and extrusion of mitochondrial DNA into the cytoplasm, which elicits a strong inflammatory response. Additionally, plastic debris promotes senescence of endothelial cells through oxidative stress, accompanied by increased expression of nicotinamide adenine dinucleotide phosphate oxidase and downregulation of sirtuin, a DAD+ dependent deacetylase that regulates several processes by modifying gene expression through histone deacetylation [78]. Plastic particles also promote premature senescence in two human lung-related cell lines (A549 and BEAS-2B) as well as in the mouse lung by deregulating the balance between intracellular reactive oxygen species and the antioxidant system of lung cells [79]. Additionally, senescence has been shown in the cultured alveolar epithelial cell line MLE12 and in rat lung cells following microplastic treatment, involving a molecular mechanism where circ_kif2cb promotes the expression of non-coding RNAs, the miR-346-3p [80]. On the other hand, plastic debris has also been revealed to induce senescence in cultured rat renal interstitial fibroblasts (NRK-49F cells), epithelial cells from human proximal tubular (HK-2 cells), and in nephrocytes of mice via the klotho/Wnt/β-catenin signaling pathway [81]. Moreover, senescence following exposure to plastic fragments has been reported in cultured C17.2 neuronal progenitor cells derived from the embryonic mouse cerebellum, as well as in proliferative neural stem cells located in the adult mouse hippocampus. In both cell types, microplastics increased the expression of the cell cycle arrest markers, p16, p21, and p27, while decreasing cyclin D expression. In both proliferating cell types, plastic particles produce cellular senescence by triggering cell cycle arrest at the G_1_ phase through mitochondrial dysfunction [28].

Based on the above-mentioned findings, it can be concluded that exposure to microplastics and nanoplastics induces senescence in various cell types. Interestingly, the incorporation of BrdU into DNA also triggers a wide range of detrimental alterations in the DNA double helix, including the induction of cellular senescence. In this context, studies have found that BrdU exposure induces senescence-like morphological changes in cultured embryonic fibroblasts as well as in immortalized human colon carcinoma cell lines (HCT116 and HCT116/80S14), even in the absence of functional p16, p21, and p53 [82]. In addition to these phenotypic alterations, BrdU treatment upregulates the expression of senescence-associated proteins such as p16, p21, p53, and the mortality marker mortalin. Similarly, BrdU exposure in neurosphere cultures derived from mice during the perinatal period has been shown to induce senescence in neural stem and progenitor cells, mediated by the activation of p53 and components of the retinoblastoma protein pathway [83].

Collectively, these findings demonstrate that both microplastics and BrdU induce cellular senescence. Notably, BrdU exposure has been shown to promote the expression of key senescence-associated proteins, including p16, p21, and p53 proteins that are similarly upregulated following exposure to microplastics and nanoplastics. Despite these parallels, the mechanism involved in the induction of senescence is not well understood. Some studies suggest that chromatin decondensation, driven by BrdU incorporation into scaffold or nuclear matrix attachment region sequences, may serve as an initial trigger for senescence [84]. Furthermore, two distinct mechanisms have been proposed to explain BrdU-induced senescence: (i) BrdU acts as a DNA intercalating molecule, preferentially inserting at adenine/thymidine-rich regions, thereby making genes with high A/T content particularly susceptible to its deleterious effects [85], and (ii) BrdU disrupts gene expression by targeting a highly conserved domain within the N-terminal tail of histone H2B, thereby altering nucleosome positioning and chromatin structure [86].

Current results have indicated that both plastic particles and BrdU promote cellular senescence and reinforce the evidence that this halopyrimidine analog may not be a reliable marker for identifying neuroblasts undergoing DNA synthesis.

### 4.3. Plastic Particles and 5-Bromo-2′-Deoxyuridine Alter DNA Methylation

In eukaryotes, methylation of DNA is a chemical modification in which methyl groups are added to the DNA molecule. Specifically, a methyl group is transferred to the 5-carbon position of a cytosine to form 5-methylcytosine. This chemical reaction is regulated by a family of enzymes known as DNA methyltransferases [87]. DNA methylation plays a crucial role as an epigenetic mechanism by regulating gene expression, primarily through the inhibition of transcription factors binding to DNA. This process is involved in several biological processes, including X-chromosome inactivation, chromatin structure regulation, neuronal activity, DNA repair, and the maintenance of cell identity [88].

Microplastics and nanoplastics can cross the nuclear envelope and directly interact with DNA, leading to damage. Previous studies have reported DNA degradation following microplastics exposure in mussels [89] and demersal fish species [90]. Additionally, evidence suggests that plastic particle exposure may alter DNA methylation patterns, although the direction and extent of these changes appear to be species-dependent [91]. For instance, DNA hypomethylation has been seen in zebrafish following microplastics exposure [92], with similar findings reported in mussels [93]. In contrast, an increase in DNA methylation has been reported in the blood cells of rats, with the degree of methylation rising in a dose-dependent manner after microplastics exposure [94]. A proposed mechanism suggests that DNA oxidation can promote DNA methylation through the polymerase ß-DNMTs 3b during base excision repair [95].

The halogenated nucleotide BrdU is known to induce DNA demethylation. The mechanisms by which BrdU affects DNA methylation are thought to resemble those of 5-aza-2′-deoxycytidine, a DNA demethylating agent. In this context, it has been proposed that BrdU produces DNA demethylation by reducing the expression of methyltransferases [96]. Although the precise mechanism remains unclear, the distinct chemical structure of BrdU compared to endogenous thymidine likely plays a key role. Specifically, this synthetic halogenated pyrimidine incorporates a bromine atom (Br) into replicating DNA when bromouracil is substituted for thymidine during DNA synthesis [18]. Moreover, the addition of exogenous BrdU can disrupt the cellular nucleotide pool. When BrdU levels are excessive, or when the ratio of deoxycytidine triphosphate (dCTP) to BrdU triphosphate decreases, the conversion of nucleotide triphosphates to deoxynucleotide triphosphates becomes inhibited. Under these conditions of nucleotide pool imbalance, BrdU can be incorporated into DNA opposite guanine, rather than its typical pairing with adenine [97]. Based on these findings, it is reasonable to assume that genes containing bromosubstituted DNA may be transcribed incorrectly into RNA, ultimately leading to the production of defective or altered proteins, including the DNA methyltransferases.

Another mechanism involved in DNA demethylation following BrdU exposure is related to nucleosome destabilization, which results from modifications in heterochromatin organization and gene expression [86]. These changes can create conditions in which BrdU is incorporated into CpG islands–DNA regions where a cytosine is followed by a guanine–substituting for cytidine and thereby leading to the loss of CpG methylation. Interestingly, other thymidine analogs, such as CldU and IdU, have also been shown to induce DNA demethylation [96].

Again, these results support the notion that BrdU labeling after plastics particle exposure may introduce artifacts that could lead to misinterpretation of experimental data. In the context of neurogenesis, an important issue is to have a confident identification of those cell precursors engaged in the S-phase of the cell cycle. BrdU tagging is widely used to evidence cell proliferation and neurogenesis in the developing and adult nervous system. Methodological problems with BrdU-labeling can be overcome through co-labeling BrdU with cell cycle markers (PCNA, mini-chromosome maintenance protein-2, Ki67 and phosphohistone-H3) or immature neuron markers (polysialylated neuronal cell adhesion molecule and doublecortin). Another approach is the use of intracranial injection of retroviral vectors to tag proliferating neuronal precursors [98,99].

## 5. Conclusions and Future Perspectives

This review outlines the detrimental effects of microplastic and nanoplastic exposure on the central nervous system, with a particular focus on their impact on neurogenesis during both prenatal and postnatal development. This is a critical concern, as plastic particles are capable of crossing both the placental barrier and the blood–brain barrier. This report also has implications for interpreting the effects of plastic debris exposure on neurogenesis when BrdU is used as a marker to identify neuron precursors engaged in the S-phase of the cell cycle. This exogenous nucleoside has yielded valuable insights into central nervous system development under various experimental approaches. Nevertheless, the impact of BrdU incorporation into DNA is often overlooked. This is an important issue when a single high dose or repeated doses of this thymidine analog are administered. The controversy arises when BrdU-positive neuroblasts can be attributed to processes unrelated to cell division, such as apoptosis, cellular senescence, or alterations in DNA methylation. Notably, these cellular events also occur following exposure to microplastics and nanoplastics. Therefore, data obtained using BrdU should be interpreted with caution, and appropriate controls are essential to ensure that BrdU labeling accurately reflects the fraction of neuroblasts engaged in DNA synthesis. Based on previous reports [73,74], it is proposed that, to label proliferating neuroblasts, a single BrdU pulse at a dose below 100 µg/g body weight should be administered.

## Figures and Tables

**Figure 1 ijms-26-07845-f001:**
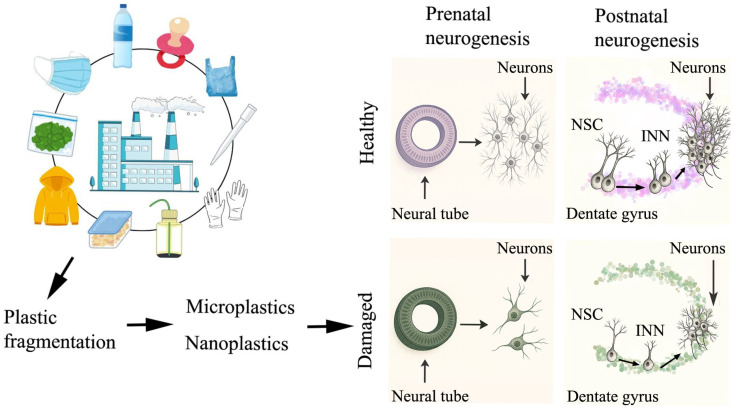
Plastic fragmentation generates microplastics and nanoplastics. These pollutants exert deleterious effects on both embryonic and adult neurogenesis. NSC: neural stem cells. INN: immature newborn neurons.

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
