# Peer review of "Effects of Microplastics and Nanoplastics Exposure on Neurogenesis: Are Thymidine Analogs a Good Option to Study Such Effects?"

_ijms, 2025, doi:10.3390/ijms26167845_

Round 1

Reviewer 1 Report

Comments and Suggestions for Authors

Dear authors,

The manuscript, titled “Effects of microplastics and nanoplastics exposure on neurogenesis. Are thymidine analogues a good option to study such effects?”, addressed the harmful consequences of plastic exposure on the nervous system, the impact of microplastics and nanoplastics on the genesis of neurons in both the embryonic period and adulthood, and the reliability of 5-bromo-2'-deoxyuridine (BrdU) labeling as a tool to analyze the effect of exposure to microplastics and nanoplastics on the proliferative behavior of neuronal precursors.

The manuscript contains a figure and describes what the literature shows about the effect of microplastic/nanoplastic on neurogenesis.

It's unclear what are the future perspectives. Please, rewrite the chapter “5 Conclusions and Future Perspectives” to improve readers' understanding.

Author Response

Response to Reviewer 1

We sincerely thank you for reviewing our manuscript. Your constructive suggestions have been highly valuable, and the following points have been modified in accordance with your recommendations. We believe that we have now addressed your comments clearly. Modifications are highlighted in blue.

The reviewer wrote (in bold and italics):

The manuscript, titled “Effects of microplastics and nanoplastics exposure on neurogenesis. Are thymidine analogues a good option to study such effects?”, addressed the harmful consequences of plastic exposure on the nervous system, the impact of microplastics and nanoplastics on the genesis of neurons in both the embryonic period and adulthood, and the reliability of 5-bromo-2'-deoxyuridine (BrdU) labeling as a tool to analyze the effect of exposure to microplastics and nanoplastics on the proliferative behavior of neuronal precursors.

The manuscript contains a figure and describes what the literature shows about the effect of microplastic/nanoplastic on neurogenesis.

It's unclear what are the future perspectives. Please, rewrite the chapter “5 Conclusions and Future Perspectives” to improve readers' understanding.

We apologize if the first version was not sufficiently clear. Chapter 5, Conclusions and Future Perspectives, has been rewritten (see pages 11-12, section entitled 5. Conclusions and Future Perspectives, lines 445–461).

Response to Reviewer 2

We sincerely thank you for reviewing our manuscript. Your constructive suggestions have been highly valuable, and the following points have been modified in accordance with your recommendations. We believe that we have now addressed your comments clearly. Modifications are highlighted in yellow.

The reviewer wrote (in bold and italics):

Thank you for your work. The information you have presented in the manuscript is helpful and interesting. The purpose of the work is very relevant. The English is fine and does not require any improvements. There are few comments on the article. In my mind, the Figure 1 is not informative enough. I suggest you to present two right pictures of brain and embryos in their normal form and in unhealthy form due to nanoplastic exposure.

We appreciate this comment and fully agree. In accordance with the reviewer’s suggestions, we have provided a new Figure 1 (see page 6, section entitled 3. Detrimental consequences of Plastic Particle Exposure on the Production of Neurons both in the Prenatal and Adult Life, lines 221-223).

In the Title of your work you say the phrase: "Are thymidine analogues a good option to study such effects"? In fact, the article talks only about BrdU. According to previous investigations [74,75] it has been reported that a single administration of 100-300 μg BrdU per 1.0 g of tissue leads to the activation of apoptotic events in the rat embryonic cerebellum. The term "administration" is incorrect in this context.

We are grateful for this comment. The term “administration” has been replaced with “injection” (see page 8, section entitled 4.1. Plastic Particles and 5-bromo-2´-deoxyuridine Induce Apoptosis, line 317).

 In the Conclusion you write that the BrdU labeling presents disadvantages for the interpretation of data related to cell proliferation and lineage tracing. Therefore, results obtained using this thymidine analogue should be carefully analyzed, and appropriate controls should be implemented to ensure that the incorporation of this marker into DNA reflects the effects of plastic particle exposure on the genesis of neurons during the prenatal and postnatal development of the CNS. This sentence is too long. Try to avoid long sentences throug the text, please. Yes. I think that it is impossible to speak unequivocally either for or against this marker. But how about concentrations lower than 100 μg/g. I think, that the conclusion must be improved and concretize. The neganive as well as the positive properties of such thymedine analogue should be described in the Conclusion. Also the different points of view should be provided in the Conclusion.

We apologize if the first version was not sufficiently clear. Chapter 5, Conclusions and Future Perspectives, has been rewritten (see page 11, section entitled 5. Conclusions and Future Perspectives, lines 444–461).

I kindly recommend you to include the short Section into your article with the information about other analogues of thymedine

We appreciate this comment and fully agree. Information about other thymidine analogues has been provided (see page 3, section entitled 1. Introduction, lines 96 – 103.

Reviewer 2 Report

Comments and Suggestions for Authors

Dear authors,

Thank you for your work. The information you have presented in the manuscript is helpful and interesting. The purpose of the work is very relevant. The English is fine and does not require any improvements. There are few comments on the article. In my mind, the Figure 1 is not informative enough. I suggest you to present two right pictures of brain and embryos in their normal form and in unhealthy form due to nanoplastic exposure. In the Title of your work you say the phrase: "Are thymidine analogues a good option to study such effects"? In fact, the article talks only about BrdU. According to previous investigations [74,75] it has been reported that a single administration of 100-300 μg BrdU per 1.0 g of tissue leads to the activation of apoptotic events in the rat embryonic cerebellum. The term "administration" is incorrect in this context. In the Conclusion you write that the BrdU labeling presents disadvantages for the interpretation of data related to cell proliferation and lineage tracing. Therefore, results obtained using this thymidine analogue should be carefully analyzed, and appropriate controls should be implemented to ensure that the incorporation of this marker into DNA reflects the effects of plastic particle exposure on the genesis of neurons during the prenatal and postnatal development of the CNS. This sentence is too long. Try to avoid long sentences throug the text, please. Yes. I think that it is impossible to speak unequivocally either for or against this marker. But how about concentrations lower than 100 μg/g. I think, that the conclusion must be improved and concretize. The neganive as well as the positive properties of such thymedine analogue should be described in the Conclusion. Also the different points of view should be provided in the Conclusion. I kindly recommend you to include the short Section into your article with the information about other analogues of thymedine.

With the best wishes,

Reviewer

Author Response

(The authors gave the same response as above.)
